# Towards Gas Discrimination and Mapping in Emergency Response Scenarios Using a Mobile Robot with an Electronic Nose

**DOI:** 10.3390/s19030685

**Published:** 2019-02-07

**Authors:** Han Fan, Victor Hernandez Bennetts, Erik Schaffernicht, Achim J. Lilienthal

**Affiliations:** Mobile Robotics & Olfaction Lab, AASS Research Center, School of Science and Technology, Örebro University, 702 81 Örebro, Sweden; victor.hernandez@oru.se (V.H.B.); erik.schaffernicht@oru.se (E.S.); achim.lilienthal@oru.se (A.J.L.)

**Keywords:** gas discrimination, gas distribution mapping, emergency response, mobile robotics olfaction, unsupervised learning, search and rescue robot

## Abstract

Emergency personnel, such as firefighters, bomb technicians, and urban search and rescue specialists, can be exposed to a variety of extreme hazards during the response to natural and human-made disasters. In many of these scenarios, a risk factor is the presence of hazardous airborne chemicals. The recent and rapid advances in robotics and sensor technologies allow emergency responders to deal with such hazards from relatively safe distances. Mobile robots with gas-sensing capabilities allow to convey useful information such as the possible source positions of different chemicals in the emergency area. However, common gas sampling procedures for laboratory use are not applicable due to the complexity of the environment and the need for fast deployment and analysis. In addition, conventional gas identification approaches, based on supervised learning, cannot handle situations when the number and identities of the present chemicals are unknown. For the purpose of emergency response, all the information concluded from the gas detection events during the robot exploration should be delivered in real time. To address these challenges, we developed an online gas-sensing system using an electronic nose. Our system can automatically perform unsupervised learning and update the discrimination model as the robot is exploring a given environment. The online gas discrimination results are further integrated with geometrical information to derive a multi-compound gas spatial distribution map. The proposed system is deployed on a robot built to operate in harsh environments for supporting fire brigades, and is validated in several different real-world experiments of discriminating and mapping multiple chemical compounds in an indoor open environment. Our results show that the proposed system achieves high accuracy in gas discrimination in an online, unsupervised, and computationally efficient manner. The subsequently created gas distribution maps accurately indicate the presence of different chemicals in the environment, which is of practical significance for emergency response.

## 1. Introduction

The environmental conditions in catastrophes can be dangerous or difficult for humans to operate since disaster sites are often structurally unstable, rubble-filled, or under low-visibility conditions. In some emergency response scenarios, the presence of hazardous airborne chemicals, such as explosive or toxic chemical compounds, can have dramatic consequences for personnel and the affected areas. Emergency responders, such as firefighters, hazardous materials and search and rescue teams, should avoid to be directly exposed to these dangers during their operation. For this reason, remotely controlled or autonomous mobile robots with gas-sensing capabilities are an important safety improvement in such disaster events, which can be deployed quickly to inspect hazardous areas without risking human personnel.

There are several requirements for using gas-sensing robots for emergency services. First of all, a gas-sensing robot should meet the demand of fast deployment. The speed and efficiency of emergency response are key, as any delay can result in negative consequences. Deploying a wireless sensor network is a solution to obtain information in the field. However, it is hard to set up a dense grid of stationary gas sensor units under time constraints. The area of interest in emergency response is usually dynamic and unstructured, which poses difficulties to deploy a wireless sensor network so as to achieve sufficient sensor node coverage and reliable data transmission. Moreover, the use of several sensor nodes has the associated challenge of calibration, which, for sensors based on electrochemical or metal oxide layer reactions, is not a trivial problem [1,2]. In comparison to a static sensor network, using a mobile robot equipped with gas sensors is advantageous. A robotic platform is more flexible as its mobility allows to perform continuous sampling in many more positions to improve the coverage of the target area. Second, a gas-sensing robot is expected to provide information related to hazardous chemicals without significant delay. Conventional techniques of analyzing chemical compounds are not developed to satisfy such demand. For example, using gas chromatography, several hours are required to identify unknown samples in the laboratory, not counting the time spent on taking and delivering the samples. Instead of analyzing the collected data offline, analysis should be carried out on-board, reliably and in real-time to allow for quick, informed conclusions on-site. To achieve such functionality, the developments in the research domain Mobile Robot Olfaction (MRO) need to be adopted to the requirements of emergency services.

MRO concerns mobile robots equipped with in-situ or remote gas sensors that carry out different tasks, including gas discrimination, gas distribution modeling, gas leak detection and localization. In particular, for applications of emergency response, gas discrimination and gas distribution modeling are of high priority. Since the presence of multiple chemical compounds is expected in many cases, the primary tasks an MRO system in an emergency response scenario should carry out includes but is not limited to (1) discriminating target compounds from interferent substances; (2) investigating how the detected chemicals are distributed in the environment; and (3) localizing the sources of target analytes if needed. The resulting information can help responders to distinguish between areas with high concentrations of harmful gases from areas that are safe to operate or evacuate [3]. In the rest part of this paper, we mainly focus on algorithmic challenges of MRO in these scenarios.

The nature of emergency services and harsh environmental conditions poses a number of challenges to fulfill the above requirements. A common issue in emergency response applications is the lack of a-priory information about the airborne substances in the environment. Regarding the aforementioned gas discrimination task, when the target compounds and interferents are fully known, a supervised gas classification system can be trained to recognize all present chemical substances. However, when the interferents are partially unknown, supervised learning is not feasible. One promising solution is to learn the discrimination model from the data collected in the field using unsupervised approaches, which is also the strategy we considered for gas discrimination in this paper. We will describe it as a part of a gas-sensing system we propose. Secondly, the harsh environments in emergency scenarios are usually demanding to both gas sensors and robotic platforms due to one or more of the following extreme conditions: high operating temperature, corrosive or explosive media, high pressure, significant vibration, high humidity, high radiation levels, electromagnetic spikes [4]. These basic issues are addressed by using robust hardwares developed to survive in harsh environments. Third, limited computational capabilities is also a considerable challenge, because robots to be used for emergency service usually are built as an overall system integrated with several computationally demanding devices, such as radar, cameras, laser scanners, communication devices, gas sensors and the corresponding software. In order to perform gas sensing online, the MRO algorithms should be adjustable to balance the computational cost and performance.

We propose a gas-sensing system for a civil robot designed to support emergency responders in an emergency situation like a house fire or gas leak. The gas-sensing capability of the mobile robot allows the responders to assess and cope with the situation without exposing themselves to danger posed by poisonous and explosive chemicals or other risk factors. Along with the gas-sensing modality, the robot is equipped with perception modalities, such as radar, cameras, laser scanners, to form a complete system. The information from different perception modalities is fused to further provide situation awareness and decision support. Our work is to address the risk factors posed by unknown gaseous chemicals in the environment, using the MRO algorithms to detect, discriminate and model the spatial distribution of detected gases. Accordingly, the gas-sensing system is divided into three subsequent modules, namely gas detection, gas discrimination and gas distribution mapping. The gas detection module is based on an ensemble one-class classifier that serves to tell gas patches from clean air, and trigger model learning in other modules. The gas discrimination module can determine the number of present chemicals fully automatically, and learn a predictive model to process instantaneous measurements online. This predictive model is used by the gas distribution mapping module, which derives an overall gas distribution model of the environment. Our main contributions are the ensemble one-class classifier for gas detection, and the event-triggered, unsupervised, online gas discrimination approach. This work aims to address fundamental open issues of performing gas-sensing tasks in emergency response scenarios, i.e., the lack of prior information, the limitation through on-board computational recourses, and the need for instant situational awareness.

The outline of the paper is as follows: in Section 2 we begin with a brief description of the development of gas sensing and robotic technologies that have benefited emergency responses in recent years. Then we review MRO algorithms that are relevant to gas sensing in emergency response applications in the fields of gas discrimination and gas distribution modeling. Next, we present our integrated gas-sensing system in Section 3. In Section 4, we describe the robotic platform, gas sensor hardware, and the specific setup of the experiments for validation. The corresponding validation results are presented in Section 5. Finally, in Section 6, we make a summary of the presented work as well as its limitations, and propose some directions for future research.

## 2. Related Work

In natural and human-made disasters, both robotics and gas-sensing techniques have been applied to protect emergency responders from dangers as well as to mitigate the corresponding damage to public health and the environment. For instance, during the Fukushima earthquake in 2011, UltraRAE 3000 benzene monitors and hand-held colorimetric detection tubes were used to detect benzene and flammable gases from the oil slick, drilling rigs and platforms [5]. Robot platforms for emergency response, including aerial, ground and maritime robots, play an increasingly crucial role, providing services such as logistics and communication support, object removal, search and rescue, etc. The earliest use of robots in an emergency response scenario perhaps is the attempt to detect radioactivity and remove debris using remote-controlled robots in the Chernobyl disaster. However, this attempt failed because the robots were unable to function due to the high radiation. A similar job was successfully accomplished by robots developed to operate post-disaster recovery missions for the Three Mile Island nuclear accident. A more time-critical use case is that some tethered and wirelessly operated robots were deployed to explore the wreckage beneath the rubble of the collapsed World Trade Center after the 9/11 terrorist attacks in New York. Since then, there have been around 50 documented robot deployments at disasters around the world [6]. More recently, many research projects have been conducted to enhance the utility of robots in a various post-disaster recovery, search and rescue operations. A noticeable example is the collaborative European project Integrated Components for Assisted Rescue and Unmanned Search operations (ICARUS). In this project, aerial, marine and ground-based robots are developed for a diversity of emergency response missions in urban and maritime environments, such as detecting victims using thermal imaging sensors, delivery rescue equipment, and marine salvage [7]. However, to the best of our knowledge, robots that are built particularly for carrying out gas-sensing tasks in emergency response scenarios are rarely reported. A recent example is a search and rescue robot for underground coal mine environments proposed in [8]. This robot can provide detection of methane at adjustable heights, but its gas-sensing capability has not been further supported with MRO algorithms that would allow the robot to perform complex tasks.

Mobile robots with gas-sensing capabilities have been already put into practice as commercial products or research prototypes. For example, General Electric has proposed a solution that equips their robotic platforms with a suction tube which can continuous sample ambient air from the inspection location [9]. The sampled gas is fed through a Simtronics gas detector GD10P that measures the concentration of hydrocarbons. Using such remote inspection robots in confined spaces such as pressure vessels can consistently monitor the explosive atmosphere. A highlight of this gas-sensing robotic system is that its gas detector compensates for aging effects and dirt on the optics during an initial calibration, and no further calibration is required afterwards. Another notable feature is the gas sensor controlled switch that shuts down the robot and other devices connected to it when observing 10% of the lower explosion limit in order to prevent the robot from igniting a flammable atmosphere. Similar feature can be found in the RoboGasInspector, which is a robot system for gas leakage detection in industrial facilities [10]. The gas-sensing payload of the RoboGasInspector includes a remote methane leak detector, and an optical thermal imaging camera to visualize gases of relatively high concentrations. Although the above two examples are robots operating in industrial sites, their gas-sensing-based security mechanism may also be useful in emergency response scenarios. Moreover, various researches are made to add more intelligence on the olfaction robots, that is to say, to perform more sophisticated gas-sensing tasks such as gas discrimination, gas distribution mapping, gas source localization & declaration and sensor planning. In the rest of this section, we cover previous work on gas discrimination and gas distribution modeling, since the corresponding functionalities are important for emergency response.

### 2.1. Gas Discrimination

Gas discrimination with mobile robots are usually performed with an electronic nose (e-nose), which is a device that combines arrays of partially selective gas sensors and pattern recognition algorithms [11]. Although the selection of the sensors is application specific, metal oxide (MOX) sensors are the most common choice for e-noses because of their wide commercial availability, fast response times and high sensitivity to the compounds of interest [12]. In the scope of this work, we are interested in the works on gas discrimination using MOX-based e-noses.

Trincavelli and co-authors [13] carried out a thorough evaluation on different classification algorithms and feature extraction approaches. Mobile robots were equipped with an e-nose and used in diverse experiments, which included a wide range of scenarios, such as a robot arena (i.e., a closed room), indoor corridors and an outdoor courtyard with mock up gas sources. The authors point out that, in such real-world environmental conditions, the steady state in the sensor response time series is rarely reached due to rapidly fluctuating gas concentration levels. Therefore, the analysis should be based on the transient phase of the signal. Since only passing the transient phase to the predictive model neglects information from other response phases, some other works consider the whole signal in the discrimination process. For example, Schleif et al. [14] proposed an approach based on Generative Topographic Mapping Through Time (GTM-TT) that can deal with high-dimensional and short temporal sequences data, which allows the robot to perform rapid classification using short data sequences [14]. In addition, the authors observed that using the data from transient phases can improve the classification performance. The work reported by Hernandez et al. [15] addresses a common issue with data in robotic olfaction applications, which is unbalanced datasets with respect to concentration levels. Such unbalanced datasets have very few high concentration measurements, while diluted measurements are present most of the time. The authors consider all phases in the signal, and integrate concentration levels into the class posterior estimation to reflect class separability. More recently, Monroy et al. [16] investigated the role of motion speed of the sampling platform in gas classification in open environments. The authors found that the classification accuracy is negatively correlated with the motion speed. Furthermore, in order to increase the performance of classification at different speeds, a supervised classifier should be trained with data collected at similar speeds. This is an important insight for using supervised learning methods with mobile robotic platforms.

The applicability of the supervised methods introduced above can be limited when prior information of the analytes present is lacking. When sufficient training data is not available, unsupervised gas discrimination has to be applied. A recent example is in [17], where the authors used a cluster analysis based approach proposed in [18], to perform gas discrimination on the data collected by a mobile robot using an e-nose.

### 2.2. Gas Distribution Modeling

Both in-situ and remote gas sensing have been considered to support gas distribution modeling in MRO in the literature [19,20,21]. Remote gas sensors, for example, tunable diode laser absorption spectroscopy based sensors, becomes appealing as they allow for taking measurements up to considerable distances, e.g., 30 m [22,23]. However, in some emergency response scenarios, the environments are full of smoke, and therefore, the applicability of remote gas sensors is limited. In addition, such remote gas sensors are manufactured to target particular analytes. High selectivity to particular gases, however, is not always an advantage in emergency response applications because the chemicals of interest in the field might be unknown before inspection. For this reason, in this paper, we are interested in gas distribution mapping using in-situ e-nose sensors. Since such sensors can only provide point measurements, a challenge is that the collected measurements are usually sparse due to the time constraint of exploration time. This issue is addressed by Lilienthal and Duckett in [24]. The authors proposed an extrapolation algorithm to spatially convolute the point measurements using a Gaussian kernel. This work was later extended to estimate the predictive variance, which allows to evaluate the model quality in terms of the data likelihood [25]. Another direction of improving gas distribution modeling is to consider realistic factors, such as wind vector, aging of measurements, and the presence of obstacles in the environment. Reggente et al. in [26,27] proposed to measure wind flow vector and use it to modify the shape of the Gaussian kernel in [25]. Asadi et al. presented two gas distribution modeling approaches based on the finding that recent measurements are more informative to estimate the current gas distribution [28]. The proposed approaches create time-dependent models by introducing sub-sampling or recency weights that relate measurements and prediction time. In [29] reported by Monroy et al., the temporal factor is accounted for by assigning each measurement a time-decreasing weight. The authors also considered the influence of obstacles in spatial gas distribution by modeling the correlation between positions in the area using a Gaussian Markov random field. Since in some cases an individual ground mobile robot is not necessarily the only or best option for emergency response applications, some works utilizing other robotic platforms to perform gas distribution modeling also merit attention. Aerial robots are a promising platform in gas-sensing tasks in emergency response scenarios, because they have superior mobility that can overcome environmental restrictions compared to ground robots [30]. The work of Ishida [31] is an early example of deploying an aerial robot to perform gas distribution mapping task using an e-nose. Later, Neumann and co-authors presented work considering a drone that samples gas measurements adaptively for gas distribution mapping [32]. Both works demonstrated that aerial robots are capable of mapping three-dimensional gas distribution, which offers a significant advantage of dealing with gas accumulation or leakage at different heights. In reality, it is possible that the analytes of interest might have different specific weights, and the gas relevant incidents occur at various heights.

Another direction towards more realistic and complex applications is to map multiple gaseous compounds in the environment, which requires to incorporate a gas discrimination approach with a gas distribution modeling algorithm. An early example is presented by Loutfi et al. in [33]. In this work, the substances are identified first by a supervised gas discrimination method using transient responses. Then, the identification information is considered by a gas concentration distribution mapping algorithm that does not assume the presence of only one type of gas. This work was further improved by Hernandez et al. in [34]. The authors integrated another supervised learning approach that does not discard low concentration measurements so that they can convey useful information such as indicating the absence of the target gases in the environment. The produced gas concentration maps are calibrated for each identified gas using a photoionization detector (PID), which is also a in-situ gas sensor. In addition, the variance of the concentration levels and class posterior provided by the classification system are also modeled, which can be more informative than gas concentration maps in some cases [35]. However, there are some common shortcomings shared by [33,34] that prevent them from providing fast situation awareness in practical emergency response scenarios. First, in both works, gas discrimination methods are trained before deployment under the assumption that the identities of the chemical compounds to detect are known, but this assumption typically does not hold in emergency response applications. Second, although the supervised gas discrimination methods used in their works can make predictions online once they are trained, their mapping algorithms are still used in an offline manner, which can only provide information after the robot exploration process is finished. In the next section, we introduce our contributions to overcome these limitations.

## 3. Online Gas Sensing in Emergency Scenarios

In this section, we present a gas-sensing system for mobile robots using an e-nose. This system follows a three-stage structure as illustrated in Figure 1. During the whole process of robot exploration, the predictive models in each module are dynamically updated to reflect newly acquired information. The gas-sensing process begins by detecting the presence of the chemical substances from instantaneous responses of a partially selective array of sensors. As mentioned earlier, the identities of the chemical substances are often unknown in emergency response scenarios. The following procedure after detecting gases is thus carried out to distinguish between different substances. To achieve this, the gas discrimination module learns an unsupervised predictive model (a probabilistic classifier) from the collected measurements and uses it to classify previously acquired and incoming measurements until the discrimination model needs to be updated. Next, the measurements are assigned to a class posteriors obtained from the probabilistic classifier, and coupled with their corresponding coordinates. This information is then fed to a gas distribution algorithm to generate spatial gas distribution maps for each chemical substance. In the following sections, we introduce each of the three modules in detail.

### 3.1. Gas Detection Using Ensemble One-Class Classifiers

Differentiating chemical analytes from the reference gas (usually clean air in open environments) is a fundamental step in gas-sensing tasks. A typical strategy of gas detection is to rely on a dedicated gas detector, e.g., PID, and use pre-defined safety thresholds. However, in the scope of this work, accurate concentration estimation in absolute gas concentration units, e.g., ppm, is not feasible because the information of the present chemicals may not be known so that the user cannot configure the corresponding gas detectors in advance. Instead, the gas detection module aims to find specific patterns in instantaneous response caused by the presence of gas using an e-nose [36]. We are aware that the baseline responses of MOX sensors might drift over time during the sampling process in open environments, which would present an unseen pattern for the same substance. Therefore, our gas detection method also recognizes drifted baseline responses in order to estimate and compensate for sensor drift.

Our gas-detection approach is twofold: first, a One-Class Gaussian Model (OCGM) is trained with baseline responses in the beginning phase of a gas-sensing task. This training phase is set to take place in the first stage of the deployment, when the e-nose is guaranteed to interact with clean air only. After the baseline learning period, the OCGM is used to predict upcoming measurements. When measurements of chemical compounds are detected, a One-Class Nearest Neighbor classifier (OCNN) is learned from these measurements. The OCNN, together with the OCGM, form up an ensemble classification system to identify the sensors’ baseline.

#### 3.1.1. One-Class Gaussian Model

A typical one-class Gaussian assumes that the data of the target class form a multivariate Gaussian distribution [37]. For a given *n*-dimensional measurement r, its probability of belonging to the target class can be estimated with the probability density function (PDF) of the Gaussian distribution, given by
(1)$$P\left(r\right)=\frac{1}{{\left(2\pi \right)}^{n/2}\left({\left|\Sigma \right|}^{1/2}\right)}e^{-\frac{{\left(r-\mu \right)}^{T}\Sigma^{-1}\left(r-\mu \right)}{2}},$$
where the parameters mean μ and co-variance Σ can be estimated from the training set. However, this Gaussian model only holds under the strong assumption that the data is unimodal, which usually is not the case for modeling response patterns of a MOX sensor array. To obtain a more flexible density model, we simplify the above multivariate Gaussian model into a linear combination of several equally weighted single Gaussians [38], as follows:
(2a)$$s_{GM}=\frac{1}{n}\sum_{j=1}^{n}\left(1-\frac{1}{n}\int_{0}^{r^{j}}P\left(r\right)dr\right),$$
(2b)$$P\left(r^{j}\right)=\frac{1}{\sqrt{2\pi {\left(\alpha \right)}^{2}}}e^{-\frac{{\left(r^{j}-\mu \right)}^{2}}{2{\left(\alpha \right)}^{2}}},$$
where $p(r^{j})$ is the PDF of the distribution estimated by the *j*th sensor response. α is a free parameter that determines the boundaries of the Gaussian model for each sensor. Intuitively, the training data, i.e., the ground truth of baseline responses, are expected to have absolute high values of $s_{GM}$. This expectation is used as a guideline to set α, such as fitting a α so that $s_{GM}>0.995$ holds for all baseline responses in the training set. A simplistic way that gives a similar result is to set α approximately 150 times larger than the range of the sampled baseline responses. The change from Equation (Equation 1) to Equations ([Disp-formula FD2a-sensors-19-00685]) and ([Disp-formula FD2b-sensors-19-00685]) avoids the Gaussian model being shaped by the co-variance. When the sensors are exposed to only diluted measurements, the cross sensitivity between each sensors can hardly be measured, which means the co-variance is not representative enough to quantify the correlation between each variable (the sensitivity of each sensor). $s_{GM}$ can indicate the likelihood that a measurement belongs to baseline responses. A rule of thumb to classify a measurement r as belonging to the target class can be derived as
(3)$$\left\{\begin{array}{cc} r\in X, & \mathrm{if}\;s_{GM}\left(r\right)>\theta_{G}\cdot \underset{r_{i}\in X}{\min}\left(s_{GM}\left(r_{1}\right),s_{GM}\left(r_{2}\right),\ldots ,s_{GM}\left(r_{i}\right),\ldots \right)\\ r\notin X, & \mathrm{otherwise} \end{array}\right.$$
where $X=[r_{1},r_{2},\ldots ,r_{i},\ldots ]$ are the data of the target class, and $\theta_{G}$ serves as a threshold to separate baseline responses and responses of chemical analytes. Since α is set to let all measurements in the training set have $s_{GM}>0.995$, as a heuristic guideline, we usually set $\theta_{G}\approx 0.995$.

#### 3.1.2. One-Class Nearest Neighbor Classifier

The One-Class Nearest Neighbor (OCNN) classifier we use is modified from a two-layer-neighborhood one-class classifier proposed in [39], which is learned by considering only positive examples as the target class. It determines the membership of a measurement r by the following rule:(4)$$\frac{\frac{1}{J}\sum_{y\in NN(r)}^{J}d\left(r,y\right)}{\frac{1}{J\cdot K}\sum_{y\in NN(r)}^{J}\sum_{z\in NN(y)}^{K}d\left(z,NN\left(z\right)\right)}<\theta_{NN},$$
where d(A,B) is the pairwise distance between data point *A* and *B* in feature space. y is one of the *J* nearest neighbors of r, and NN(y) is the set of *K* nearest neighbors of y. $\theta_{NN}$ is a threshold parameter that can be learned by cross-validation. Figure 2 depicts the consideration of two-layer-neighborhood.

The implementation of the proposed OCNN is shown in Algorithm 1.
**Algorithm 1** One-class nearest neighbor classifier**Input:** the training data of the target class X, the test sample r**Output:** the OCNN score of the test sample $s_{NN}$1:Compute the distances between r and its *J* nearest neighbors in X, and find the median value $M_{r}$2:**for all** nearest neighbors y∈NN(r)
**do**3:  Compute the distances between y and its *K* nearest neighbors NN(y) in X, and find the median $M_{y}$4:  **if**
$M_{r}$<$M_{y}$
**then**5:     $s_{NN}\leftarrow \left(s_{NN}+1\right)/J$.6:  **end if**7:**end for**8:**return**$s_{NN}$

#### 3.1.3. Ensemble One-Class Classification System

Creating an ensemble of diverse classifiers is a way to improve predictive performance [40]. The proposed one-class classification system for gas detection consists of OCGM and OCNN described above. More like boosting instead of typical ensemble methods, in our proposed approach, the training phases of the two classifiers take place one after another, and the positive examples for training are from different classes even though the ensemble classifier only targets the baseline responses. The training procedure of our approach is as follows:Collect an amount of baseline responses B as the training data for the OCGM classifier.The parameters μ, σ, and α in Equations ([Disp-formula FD2a-sensors-19-00685]) and ([Disp-formula FD2b-sensors-19-00685]) are learned from B.Apply the trained OCGM to process instantaneous measurements online.The measurements that satisfy the condition $s_{GM}<\theta_{GM}$ are labeled to be not baseline responses, and they are stored as a training set C for the OCNN classifier.An OCNN classifier is trained with C according to Algorithm 1.Use the OCNN classifier to process the data in B, and find the mean $\mu_{NN}$ and variance $\sigma_{NN}$ of the outputted $s_{NN}$ scores. Accordingly, a rule of thumb for identifying baseline responses is introduced as follows: if the $s_{NN}$ score of a given measurement falls inside the interval $\left[\mu_{NN}-3\sigma_{NN},\mu_{NN}+3\sigma_{NN}\right]$, then it is identified as baseline response.Both OCGM and OCNN now form up an ensemble classification system for gas detection. For a given measurement, it will be recognized as a baseline response if its $s_{G}M>\theta_{GM}$ or $s_{NN}$ falls in the interval of $\left[\mu_{NN}-3\sigma_{NN},\mu_{NN}+3\sigma_{NN}\right]$.

In practice, the initial training data B can be prepared as follows: before the sensor array is deployed into the target field, we let the sensor array only interact with the clear air for a period of time $T_{B}$. During $T_{B}$, the collected measurements are considered as the initial B. Afterwards, the gas detection model is ready to make predictions on instantaneous responses.

The gas detection procedure also plays an important role in baseline correction, which is an essential step to process instantaneous MOX sensor responses. As suggested in [41], differential baseline correction is employed here to compensate for noise, drift, and inherently large or small signals. In this method, the mean value of the baseline responses is extracted as the baseline offset. However, this offset may not be constant due to the effects of temperature, humidity and short term sensor drift, which has a negative effect on identifying the baseline response and performing gas discrimination. An example is shown in Figure 3. In this exemplary dataset, a MOX sensor was exposed to two chemical compounds separately in the period E1, E2 and was exposed to clean air in the period B0, B1, B2. As is shown, the initial baseline observed in the beginning (B0) was hardly recovered in B1 or B2. Our proposed ensemble one-class classifier allows to recognize the drifted baseline responses such as the ones in B1 and B2. Consequently, the set of baseline responses is extended by including newly recognized data, and therefore, the drifted baseline offsets can be learned (i.e., baseline modeling in Figure 1). A detailed description of when the baseline offset is updated is presented in Section 3.2.1.

### 3.2. Towards Gas Discrimination in Emergency Scenarios

In this section, we introduce the gas discrimination module that allows to perform unsupervised gas discrimination online, and to update the predictive model as more measurements are collected.

#### 3.2.1. Adaptive Model Learning

Since we do not assume full prior information about chemicals present in the environment, predictive models for gas detection as well as gas discrimination cannot be trained in advance. Instead, the information for modeling is incrementally collected as the robot carries out gas-sensing tasks. For this reason, the predictive models should be adaptive to reflect newly observed patterns. An intuitive strategy to obtain up-to-date models is to perform re-learning periodically at a given frequency or to learn a new model after a number of measurements have been acquired. However, such update strategies are likely inefficient. For example, when the sensors are only interacting with diluted measurements, including these measurements into a discrimination model for chemical analytes will not contribute to better performance but only increase computational cost. As mentioned earlier, one critical challenge of carrying out mobile olfaction tasks in emergency scenarios is the limited computational resources. In some applications, there are subsequent tasks after gas detection, such as gas discrimination, gas distribution mapping, gas source tracking, and sensor planning. Performing these tasks can be computationally expensive. However, these subsequent tasks do not need to be executed continuously as long as no new gas detection event occur. In order to save computational resources while the sensor unit is not interacting with analytes, the gas discrimination learning phase is designed to be event-triggered, which means it only takes place when chemical analytes are detected. This procedure can be suitably modeled using a state machine. As shown in Figure 4, we model the gas-sensing process as a state machine that consists of three states, namely baseline, uncertain and exposure state.

The states and corresponding transition conditions are defined as follows:The baseline state corresponds to situations where the sensors are exposed to the reference gas (e.g., clean air). This state is assumed when sufficient measurements are identified as baseline responses (using the ensemble one-class classifier introduced in Section 3.1). Measurements collected in this state are used to update the baseline offset, which can compensate for baseline drift. In order to reflect a possible drift effect in time, this update takes place every $T_{UB}$ seconds during the baseline state.In the baseline state, if gas is detected, the system will change to the uncertain state (the transition BU); if clean air measurements are acquired, the system stays in the baseline state (the transition BB).The uncertain state is an intermediate state between baseline and exposure state. This state is reached from baseline (exposure) state when baseline responses (chemical compound) are detected, the number of such measurements are not deemed to be enough to indicate a significant gas (or baseline) detection event.In the uncertain state, if clean air/gas measurements are continuously acquired, the system will move to the baseline state/exposure state (the transition UB/UE); if the continuous observation of clean air/gas is interrupted by acquiring gas/clean air measurements, the system stays in the uncertain state (the transition UU).The exposure state represents events in which analytes are detected using the ensemble one-class classifier. This state is reached when sufficient measurements have been identified as different from the baseline responses. When the state machine transits to this state, the re-learning process of the gas discrimination model is triggered to include new measurements belonging to chemical analytes. This learning process is carried out periodically, e.g., the model is re-learned every $T_{UD}$ seconds during the exposure state.In the exposure state, if clean air measurements are acquired, the system will reach the uncertain state (the transition EU); if gas is detected, the system stays in the exposure state (the transition EE).

According to the state machine described above, the baseline offset extraction only takes place during baseline states, and the model learning for gas discrimination only takes place during exposure states. This control logic for model updates allows for an efficient allocation of computational resources.

#### 3.2.2. Unsupervised Online Gas Discrimination

The gas discrimination module is based on the KmP algorithm proposed in [18], which is an unsupervised approach developed for uncontrolled environments. The original KmP algorithm is an offline learning algorithm that consists of the *K*-learning and the *m*-learning phase to learn functional parameters, and the *P*-learning phase to perform gas discrimination. In order to adapt the KmP algorithm to perform online, the three learning phases are integrated with the adaptive model learning process described in Section 3.2.1. As shown in Figure 5, firstly, the *m*-learning phase learns a concentration threshold from the adaptive baseline model. In each discrimination process, the measurements with concentration level above this threshold ($R_{E}$ in Figure 5) are then extracted to be processed in the next learning phase. In the *K*-learning phase, the number of classes of the detected analytes ($K^{*}$) is determined fully automatically. $K^{*}$ itself is an important information on the situation, and it is a crucial parameter for the subsequent *P*-learning phase, in which the extracted measurements are clustered into $K^{*}$ classes. The labeled data are then used to train a probabilistic classifier, which assigns posteriors for all the acquired measurements, as well as make predictions on the incoming measurements until the classifier is updated again with the above learning process.

Besides modifying the KmP algorithm into an online version, another improvement is made by automatizing the parameter selection process in the *K*-learning phase. In the original offline KmP algorithm, $K^{*}$ is found by applying the silhouette method in a user-defined search space [42]. For example, if the user assumes there are at least two and at most five different chemical substances are present, the search space is set to be [2,5]. The absence of prior information in emergency scenarios might make it difficult for users to decide on a search space. Our solution to address this issue is to estimate search space in an unsupervised way from the data (search space determination in Figure 5). The estimation method we use is based on an analysis of a kernel matrix of the data [43]. The idea behind this method is to exploit information from the structure of the kernel matrix. Since the elements of the kernel matrix represent pairwise proximity of the data, the kernel matrix should have a block diagonal structure after the elements belonging to the same class are moved next to each other. Search space determination starts with permuting the pairwise distance matrix to a kernel matrix using:(5)$$A\left(i,j\right)=e^{-\frac{d\left(r_{i},r_{j}\right){|}^{2}}{2\sigma_{K}^{2}}},$$
where the kernel bandwidth $\sigma_{K}$ is set to be the value at 10th percentile of the total pairwise distances, based on an empirical conclusion in [44]. The method proceeds by finding the eigenvalue decomposition of the kernel matrix. Since A is a real symmetric matrix, it can be decomposed by $A=U\Lambda U^{T}$, where U is an orthogonal matrix whose columns are the eigenvectors of A, and Λ is a diagonal matrix whose entries are the eigenvalues of A. The eigenvalues and eigenvectors are then sorted in descending order to obtain a measure of compactness for each data point, given by $\gamma_{i}=\Lambda_{i}\cdot E\left(U_{i}\right)$. Next, the plot of γ is drawn, where an elbow of the curve is expected to indicate the upper bound of the search space for the considered data. The elbow is found by using the algorithm proposed in [45]. An example is shown in Figure 6, in which the dataset includes two classes of substances. The interval of the value of $K^{*}$ is estimated to be [2,4] as the elbow of the γ-plot is found at 4 (the lower bound is always 2 using evaluation criteria such as silhouette, Davies–Bouldin, or Calinski–Harabasz [18]). Finally, the silhouette method is applied to determine the parameter $K^{*}$ using the interval just found as the search space.

### 3.3. Multi-Compound Gas Distribution Mapping

In the gas distribution mapping module, class posteriors of the detected substances are interpolated into a set of classification maps. Classification maps represent a spatial distribution of the discriminated chemical substances.

Ideally, gas distribution modeling using in-situ sensors relies on the assumption that the trajectory of sensors roughly covers the area of interest [24]. However, due to the need for fast situation awareness in emergency response scenarios, gas distribution mapping should be carried out during the exploration process before full coverage of the target area is reached. The gas distribution model is updated periodically to be able to present the current knowledge based on the acquired measurements, updates of the gas discrimination model, and the newly visited sampling locations.

We use the Multi-Compound (MC) Kernel DM+V algorithm proposed in [24,34] to map the discriminated chemical substances in the environment. The Kernel DM+V algorithm can compute spatial distributions of the gases in various forms, e.g., a mean distribution map, predictive variance map, etc. In particular, using the probabilistic classifier provided by the gas discrimination module that assigns class posteriors to observed measurements, the MC DM+V algorithm can also compute classification maps. The classification maps are a set of lattice representations of the likelihood distribution of the detected chemical compounds at their respective locations. As is shown in Figure 7, once the probabilistic gas classifier is trained, the class posteriors of the measurements $R=[r_{1},r_{2},\ldots ,r_{N}]$ for each class *l* are predicted. Then the classified measurements, along with their coordinates $x_{R}$, are used to map the spatial distribution of posteriors, as proposed in [34].

The reason why we choose to map class posteriors instead of gas concentration is because accurate concentration estimation in absolute gas concentration units is not feasible without knowing the identities of each detected chemical compound. Concentration levels can be roughly derived from instantaneous responses, e.g., by taking the mean value of each sensor response, since over a certain concentration range, the logarithm of the change in resistance is linearly proportional to the logarithm of the gas concentration [46]. This quantification should be used with caution. It could mislead the emergency responders because an uncalibrated indicator such as Ic does not precisely reflect the actual concentration, which means the toxic levels of some chemicals could be underestimated. Of course, under the condition that dedicated sensing modalities are available for gas quantification, the MC Kernel DM+V algorithm can also estimate gas concentration maps simply by replacing class posterior P(L|R) with calibrated gas concentration levels in the respective equations [34].

## 4. Experimental Validation

In order to evaluate the proposed algorithmic set-up of the gas-sensing system, it was implemented on a search and rescue robot. The robot has been tested in different scenarios including a firefighting training facility (Figure 8a,b), university laboratory rooms and a basement of a public building (Figure 8d). In this paper, we present the experiment trials conducted in the basement environment.

### 4.1. The Robotic Platform

The robotic platform used is a Taurob tracker [47], which is built to help CBRN (Chemical, Biological, Radiological and Nuclear) first responders, EOD (explosive ordnance disposal) teams, fire-fighters and search & rescue team to gain first hand information in dangerous environments. The Taurob tracker offers interfaces for easy and quick integration of external sensors and measurement devices. As an instance of firefighting support, a model of the Taurob tracker, mounted with a robotic arm and a zoom camera, is used by the fire brigades in Vienna, Austria. With the goal to make a firefight support robot more intelligent and autonomous, this version of the Taurob tracker was further extended with a novel 3D radar camera, a 3D laser scanning sensor and a thermal camera (see Figure 8c) in the EU funded H2020 project “SmokeBot” (www.smokebot.eu). Fusion of radar, thermal and camera sensors allows for robust mapping and navigation under low visibility conditions, e.g., in dark or/and smoke-filled areas, as shown in Figure 8a,b. For gas sensing, the robot is equipped with an e-nose that is designed to be partially sensitive to CO and NO_2_, which are common toxic gases produced at fires.

#### The Electronic Noses

The SmokeBot platform was equipped with a prototype array composed of several gas sensors (UWAR nose) developed by Wei et al. [48].

The UWAR nose incorporates three MOX sensors coated with tin oxide (${\mathrm{SnO}}_{2}$), tungsten oxide (${\mathrm{WO}}_{3}$) and nickel oxide (NiO), respectively. During the sampling process, the ambient air is drawn into the sensor chamber through a pipe by a micro air pump. An airflow rate sensor is employed to monitor the volumetric flow rate, which is used as a reference measure to control the pump speed. The measurements are filtered by a physical filter to eliminate smoke particles, and further moved into a measurement chamber to interact with the MOX gas sensors. The raw sensor responses are generated at 100 Hz, and they are pre-processed by an on-board micro controller in real-time with digital filters, peak detection and fast Fourier transform. After these signal processing procedure, the responses are sampled at 2 Hz as the sensor output.

### 4.2. Experimental Set-Up

In total 6 experiments were conducted to validate the presented approach (see Table 1). The experimental environment is a basement with a narrow corridor connecting two rooms. Figure 8d shows one of the rooms. We made no explicit effort to regulate the environmental conditions, such as temperature, airflow, and pressure. Since the regulations did not allow the release of toxic gases into the environment, we use three kinds of commercially available liquids as analytes: ethanol (95% pure), 1-propanol (99.5% pure) and acetone (100% pure). In a previous study, the UWAR nose showed sensitivity to these chemical compounds [17].

Except from the second trial (Experiment 2), two gas sources of different kinds were placed at separated locations in the environment (“ethanol + acetone” and “ethanol + propanol”). In Experiment 2, an extra ethanol gas source was placed at another position (“ethanol + ethanol + acetone”). The gaseous compounds were released from a beaker filled with ethanol, propanol or acetone. To facilitate evaporation, a bubbler was used. Before the start of the experiment, the sensor arrays were allowed to pre-heat for a period between 10 and 30 min, which is a standard warm-up time for MOX sensors. During the sampling process, the robot was commanded to pause at several locations, including the positions near gas sources and positions that are away from gas sources.

## 5. Results

This section is devoted to validate the proposed gas-sensing system in terms of the following aspects: (1) the performance of the ensemble one-class classifier and baseline modeling (Section 5.1); (2) a comparison between the proposed online gas discrimination approach and its offline counterpart (Section 5.2); (3) the overall results of the generated multi-compound classification maps (Section 5.3).

### 5.1. The Performance of the Ensemble One-Class Classifier and Baseline Modeling

To begin with, we present an example of using the proposed state machine to model the gas-sensing process. As shown in Figure 9a, in Experiment 2, three times the exposure state was reached, which corresponds to the periods that the robot approached the two ethanol and the acetone source. As the robot moved away from the gas sources, the sensors were exposed to clean air, so the system returned to the baseline state. In Figure 9b, we visualize the sensor drift that occurred in this trial. There were 29 updates of the baseline offset in total. In each update, one can find that the median and maximum values of the baseline responses were larger than before for all three MOX sensors of the UWAR nose. This result indicates that the proposed adaptive baseline modeling procedure is necessary, and drift compensation can be conducted as long as baseline states are properly reached.

The baseline modeling relies on the performance of the ensemble one-class classifier. Figure 10a–c exemplify that using OCNN and OCGM together can accurately identify baseline responses. In each of these figures, the first box-plot shows that the $s_{NN}$ scores of all measurements in the training set C (see its definition in Section 3.1.3) range from 0 to 1. In the second box-plot, we select only measurements from the C that have $s_{GM}$ values larger than the third quartile of all $s_{GM}$ scores in C. This subset $C^{-}$ includes measurements corresponding to relatively low concentration levels, which means they are more likely to be misclassified as baseline responses by the OCGM. Nevertheless, the risk of misclassification can be reduced by monitoring $s_{NN}$ scores. One can notice that the $s_{NN}$ distribution becomes much narrower in the second box-plot of each figure. If we further look at the $s_{NN}$ scores of baseline responses (either 3rd, 4th or 5th box-plot), we find that their $s_{NN}$ values appear to be very stable, and rarely overlap with $C^{-}$. This validates that setting a proper interval of $s_{NN}$ values can identify baseline responses, and those measurements that might have $s_{NN}$ overlapping scores with baseline responses in C, can be easily filtered out.

### 5.2. Efficiency Comparison between Updating Gas Discrimination Model Online and Offline

In the previous subsection, we showed that the ensemble one-class classifier manages to support the state machine to achieve adaptive model learning. In this subsection, we show that the gas discrimination module benefits from the resulting event-trigger control logic. Compared to performing the original offline KmP algorithm [18], the proposed online gas discrimination module processes less data when updating the predictive model, because it only carries out model learning within the exposure state. In contrast, the offline algorithm has to process the instantaneous measurements of the whole time series up to the update moment, including a number of diluted, less informative measurements. In order to show the difference in the efficiency between online and offline learning process, we compare in two aspects:For each experiment trial, we compare the amount of data that had to be processed between online KmP and offline KmP (as if it is used in the way of batch learning). When the state machine reaches the exposure state, the update of the gas discrimination model begin to take place periodically. In all experiment trials, the update frequency was $T_{UD}=30$ s. We assume that the offline KmP algorithm also updates its model at the exact same times. In each update, the online model considers $n_{t}$ measurements, whereas the offline model considers $n_{E}$ measurements. The value of $\left(n_{t}-n_{E}\right)/n_{t}$ can be considered as an index of the data size reduction in the updates. In Table 2, we report the mean ($\mu_{DR}$) and standard variance $\sigma_{DR}$ of $\left(n_{t}-n_{E}\right)/n_{t}$ for each experiment trial.When the robot has visited all the gas sources, the ground-truth of $K^{*}$ is available, which can be used to examine the efficiency of the *K*-learning phase. For each trial, we compare the performance of both approaches in estimating $K^{*}$, given the data collected when the online discrimination model was last updated. As introduced in Section 3.2.2, in the *K*-learning phase of the online algorithm, the search space is learned from data at first, whereas the offline KmP algorithm uses the search space [2,5] by default [18].

The above results indicates that the size of the data to process for the online model processes can be reduced by at least 54.04% on average in the updates of a trial, compared with using the offline KmP directly. Considering that the KmP algorithm has $O(N^{2}\cdot \log N)$ complexity, halving the data to process is a significant adjustment especially in time critical application scenarios. In addition, the online approach is more accurate in estimating $K^{*}$.

The accuracy of the gas discrimination model is of great importance as the prediction of the acquired and incoming measurements can contribute to situational awareness. We calculate the accuracy of every multi-class prediction in each trial to validate the online gas distribution module. Ground-truth was obtained by manually labeling all the measurements in the exposure state. Since the sampling positions and sensor readings of these measurements are associated with time stamps, it is rather easy to determine which gas source these measurements correspond to (by checking the experiment process records). The reason that we do not consider the diluted measurements collected when the robot is far away from either gas sources is because the actual identities of these measurements should be clean air, which make them not suitable to be used in the evaluation on the discrimination between chemical substances. Since all the gas discrimination tasks during the exposure state are 2-class classification problems, the simplistic evaluation index accuracy is given by
(6)$$H=\frac{\#\mathrm{measurements}\;\mathrm{correctly}\;\mathrm{labeled}\;\mathrm{positive}+\#\mathrm{measurements}\;\mathrm{correctly}\;\mathrm{labeled}\;\mathrm{negative}\;}{\#\mathrm{total}\;\mathrm{measurements}\;\mathrm{in}\;\mathrm{exposure}\;\mathrm{states}}$$
where positive and negative class correspond to the two different chemical analytes to be discriminated. The result in Figure 11 shows that the online discrimination model is very accurate on the measurements acquired in exposure states. Please note that these accuracies are achieved by performing unsupervised learning online, and although the evaluation only considers data in exposure states, the discrimination model can also process measurements in other states, as well as incoming measurements.

### 5.3. Multi-Compound Classification Maps

As described in Section 3.3, the classification maps of clean air and each detected chemical analyte are the considered model of gas distribution. We present example results of three trials, which are representative for each type of the experimental set-up, namely “ethanol + acetone” (Experiment 1), “ethanol + propanol” (Experiment 3) and “ethanol + ethanol + acetone” (Experiment 2). Figure 12, Figure 13 and Figure 14 are snapshots of the classification maps produced when the robot has visited all gas sources in Experiment 1, Experiment 2 and Experiment 3 respectively. In the figures, the gas source positions (triangle markers), the exploration paths (yellow lines), and the way points that the robot paused for more than 2 s (green dots) are marked. The classification maps contain information that corresponds to the actual experimental set-ups, i.e., the respective gases were observed and classified close to their respective sources. In all results, one can observe that areas with high posteriors of chemical analytes lay close to the actual gas source locations. The areas predicted to have high posteriors of clean air always appear to be far away from gas sources. Since we set the default posterior to 0 in each classification map, areas away from measurements have low posteriors in all classes. This implies we do not have certain knowledge on these positions based on the current model. In Figure 12 and Figure 14, it is rather easy to estimate the approximate source positions from the high-posterior areas without further exploitation of the gas distribution models (e.g., extracting corresponding variance maps). In the more complex trial Experiment 2, in which there are three gas sources and two of them correspond to the same chemical, the classification map of the ethanol (Figure 14b) is less indicative to directly infer the two ethanol sources. Nevertheless, these classification maps are still informative because they reflect the fact that only two different chemical analytes are present in the environment. The gas discrimination module determines the correct number of chemicals ($K^{*}=2$), and suggests that the measurements collected near two ethanol sources belong to the same class. The subsequent gas distribution mapping module predicts that the area around the first two sources is very likely to be contaminated with the same chemical analyte.

## 6. Conclusions and Future Work

In this paper, we present a mobile robotic olfaction system that allows to perform online gas-sensing tasks in unknown open environments. We focus on emergency response scenarios, which always require time critical operations, while the information of the possible target chemicals, such as their numbers and identities, is often absent before deployment. In order to reduce the demand of computational power, the proposed model learning process is event-triggered, which is achieved by a state machine model and a gas detection module based on an ensemble one-class classifier. The gas detection module can recognize baseline responses under sensor drift, and therefore, contribute to compensate such drift. With the support of the gas detection module, the gas discrimination module requires significantly less computational power compared to the offline counterpart algorithm. Our unsupervised approach makes accurate prediction on the number of the chemical analytes, which provides important information that cannot be obtained from supervised approaches. We further integrate the discrimination model with a gas distribution mapping algorithm to interpret the environment in the form of classification maps. In a realistic emergency response mission, approaching the vicinity of the suspicious gases can be done with the necessary preparations using the classification maps as reference, and therefore, reduce the danger for first responders.

We are aware that in real-world emergency response applications, the situations of how many and where the chemical compounds are present can be more challenging for modeling gas distribution than in our experimental set-ups. For example, if two gas sources of different chemicals are located too close to each other, there is a chance that the measurements acquired between the sources come from a gaseous mixture composed of both chemicals, which are likely to be misclassified by the proposed clustering analysis-based gas discrimination module. Consequently, the sources cannot be reliably separated in the resulting classification maps. In our system, the separation of the sources relies on the correct gas discrimination of measurements acquired around the source locations, which requires that the fingerprints of the detected gases are separable in the feature space. For this reason, the MOX sensors are preferably exposed to clean air before interacting with another gas. In practice, when the sources are too close, the in-between area can be contaminated with a gas mixture, which limits the applicability of the proposed system. In cases where clean air can be found between the sources, there is still a limit of separation, i.e., the minimum distance between two sources of different chemicals that the proposed approach can reliably separate. Below this limit of separation, the baseline state might not be reached and hence no baseline adaption being performed. As a result, a quasi gas mixture would appear in the response pattern. The separation limits depend on several conditions, including the nature of the gas sources, the intrinsic parameters of the used MOX sensor array such as the limit of detection, the cross-sensitivity of the target analytes, and the recovery time, as well as the motion of the robot and the airflow in the field. For instance, considering the MOX sensor array as a factor, a fast sensor can recover to the baseline state quickly when exposed to the clean air, which will benefit the proposed system with a larger limit of separation than using a sensor with slow recovery time. The airflow around the sources is also a critical factor. The limit of separation varies when the robot is mostly downwind or upwind of the gas(es) in the area between sources. In our experiment trials, the minimum distance between sources of different classes was more than 6.9 m, which is safely above the limit of separation given the sensor hardware and the environmental conditions. How to estimate the approximate limit of separation for our system (or other gas discrimination and mapping approaches) given a MOX sensor array and the possible target analytes, and how to determine if the e-nose has encountered a gas mixture, are our future research questions.

Other future work would consider to improve our work in the following aspects. First, we would like to combine the proposed gas detection module with a novel change point detection algorithm TREFEX reported in [49]. Our ensemble one-class classifier is a statistical model whereas the TREFEX models the responses of MOX sensors as a hybrid exponential signal. In Figure 15, there are some measurements that we treated as clean air even though they have shown responses to acetone. The low concentration levels of these measurements proved to be difficult for our gas detection module, but their rising curves are informative enough for the TREFEX algorithm to declare a gas detection event. Second, a one-class classifier that identifies a specific detected chemcial analyte would be a great supplement to our unsupervised gas discrimination approach. Such a one-class classifier is advantageous than a multi-class discrimination model under certain circumstances. For example, when the robot repetitively interacts with one type of unknown gas in different locations, our unsupervised approach will fail to provide correct information since it cannot built a predictive model by clustering data from only one class. In contrast, a one-class classifier can confirm that only one chemical substance is present in the environment, and report negative result when an unseen chemical substance is detected so that a multi-class model can take over the discrimination process. Third, the applicability of our system can be extended for more use cases by adopting our three-module gas-sensing framework to other robotic platforms, such as drones, multi-robot system, etc. By doing so, we expect to find some additional issues (and corresponding solutions) for gas sensing in emergency response scenarios.

We also would like to address a typical but critical hurdle when developing mobile robotic systems, which is the lack of standardized test-bed. In real-world experiments, MRO systems are tested in complex environments affected by frequently and often drastically changing conditions, such as turbulence, temperature, pressure, and changing air flow. As a result, repeatability of in-field experiments is not easy to achieve. To address this issue, one strategy is to spend effort on simulating accurate artificial gas dispersion [50,51]. However, simulations of gas dispersion are far from being able to take into account all relevant aspects of a real-world experiment. For example, the actual interaction between the airborne chemical compounds and gas sensors in open environments is not considered by computer simulation so far. Instead, we expect to improve the experimental set-up. An area that is concurrently monitored during an MRO mission by a dense gas sensor network is a desirable solution as it can provide ground-truth including the gas identity, gas concentration levels at known coordinates. Such a test-bed allows a variety of gas-sensing algorithms to be validated thoroughly, and to better compare the performance of different solutions.

## Figures and Tables

**Figure 1 sensors-19-00685-f001:**
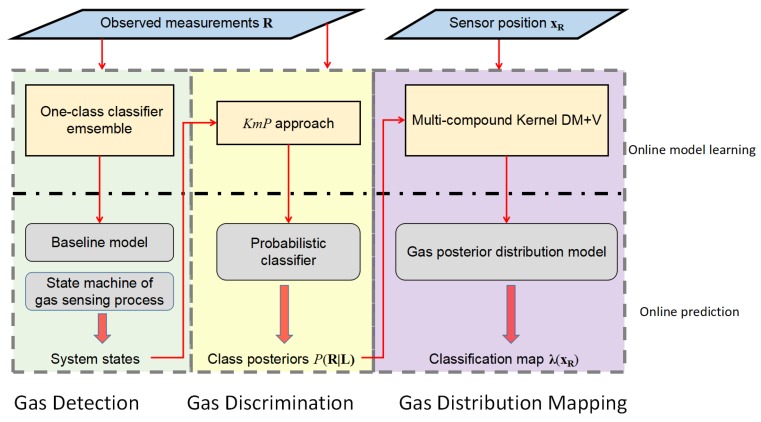
The diagram of the robotic gas-sensing system.

**Figure 2 sensors-19-00685-f002:**
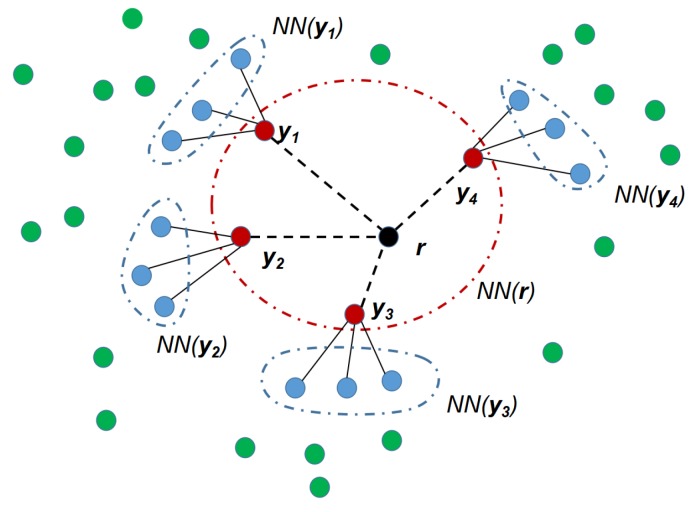
A graphical representation of considering two layers of neighborhoods for a given measurement r. The data points in red correspond to the first layer neighbors, and the data points in blue correspond to the second layer neighbors. The data points in green are outside the two layers of neighborhoods of r. In this example, the first layer was selected with neighborhood size J=4 and the second layer was selected with neighborhood size K=3.

**Figure 3 sensors-19-00685-f003:**
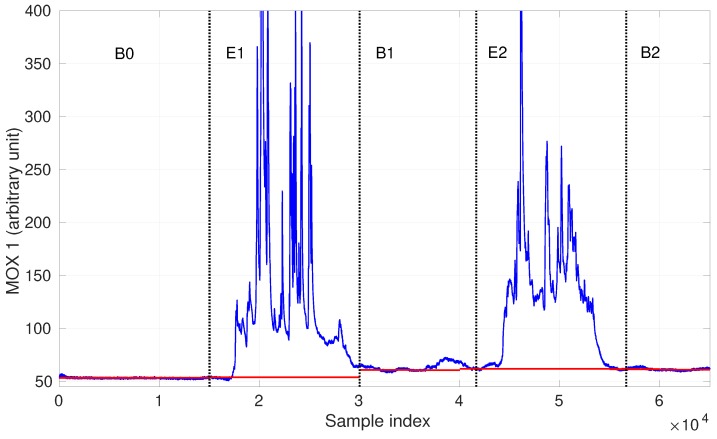
An example of baseline drift of a MOX sensor and adjusted baseline offsets (the red segments after period B0).

**Figure 4 sensors-19-00685-f004:**
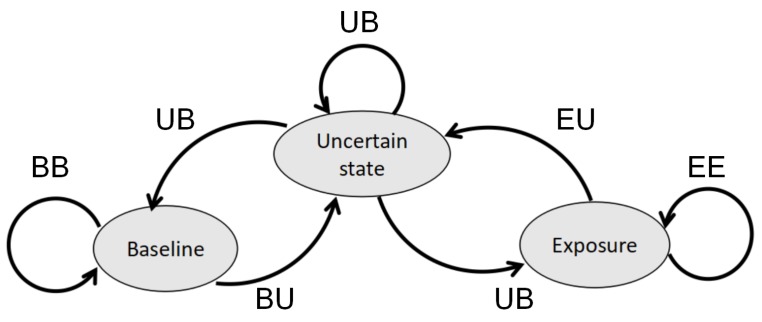
The state machine of a gas-sensing process. The transition conditions are explained further in the text.

**Figure 5 sensors-19-00685-f005:**
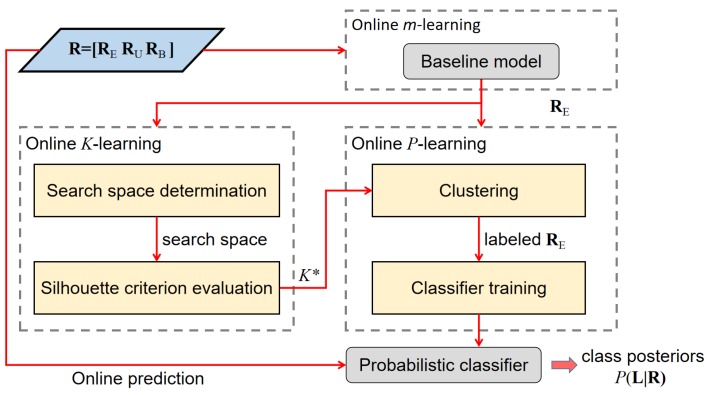
The diagram of the gas discrimination module. The learning path is triggered by the state machine, while the prediction path is used for all incoming measurements.

**Figure 6 sensors-19-00685-f006:**
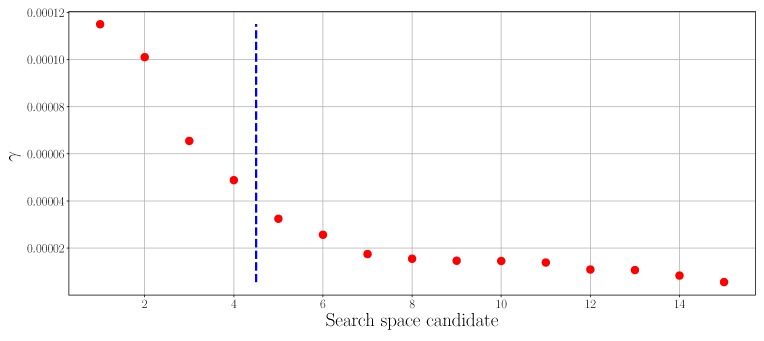
An example of determining the search space of the number of clusters. The dataset used here includes ethanol and acetone.

**Figure 7 sensors-19-00685-f007:**
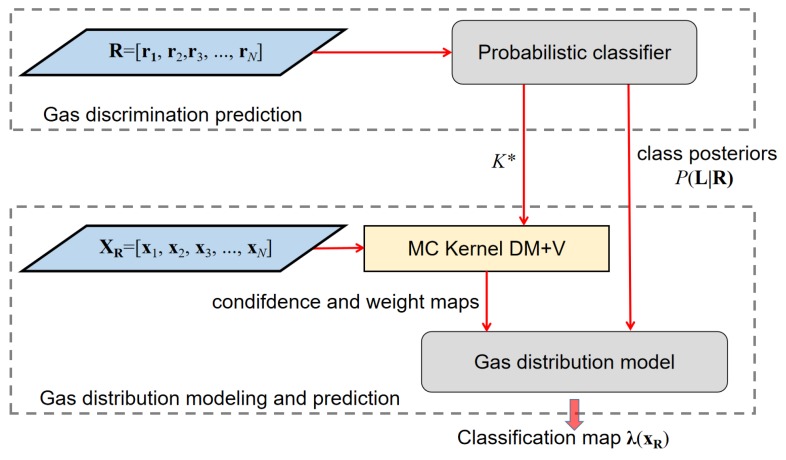
The gas distribution mapping module coupled with the class posteriors from the gas discrimination module.

**Figure 8 sensors-19-00685-f008:**
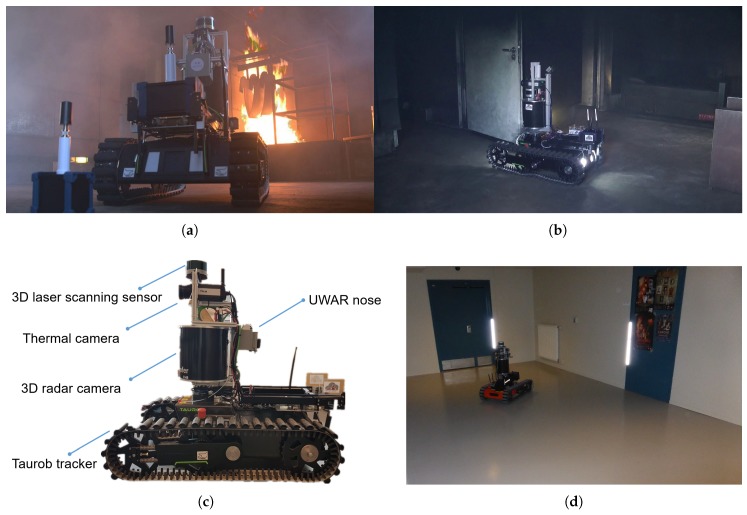
(**a**,**b**): Real-world working environments for the SmokeBot, which are imitated in a firefighter training facility; (**c**): SmokeBot sensor set-ups; (**d**): Experimental environment.

**Figure 9 sensors-19-00685-f009:**
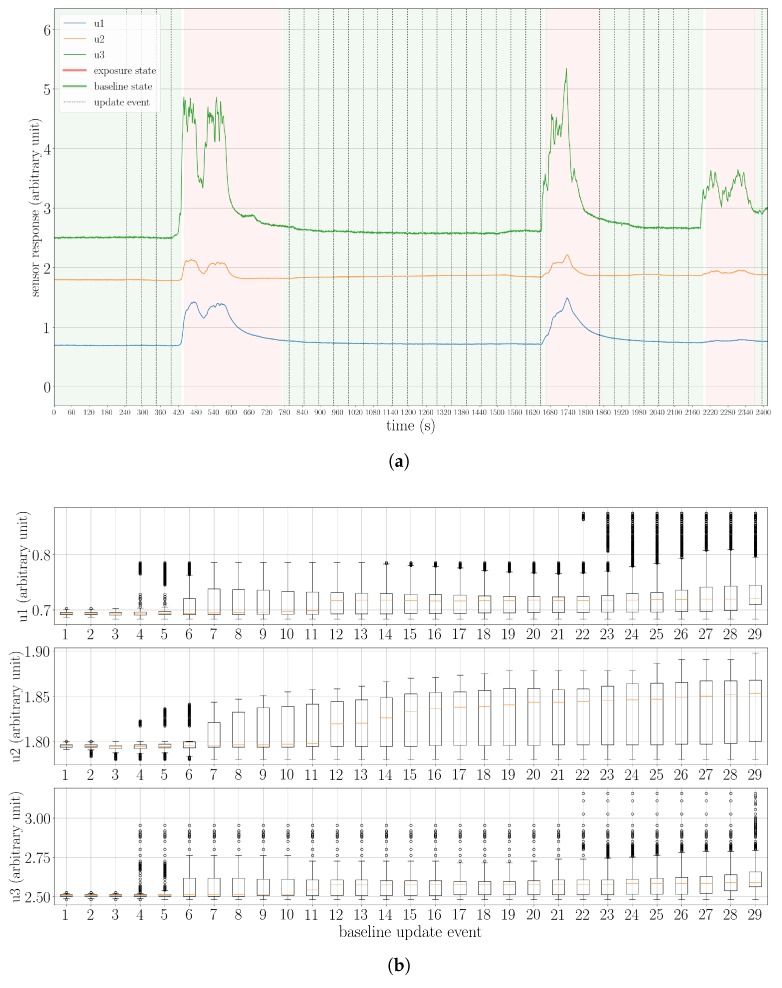
(**a**): The responses of MOX sensors and corresponding assigned states in the trial Experiment 2. In this case, the ground-truth baseline is never fully recovered; (**b**): The baseline offsets learned in each periodic updates ($T_{UB}$ = 50 s) during the baseline states. The drift effect is observable for all three MOX sensors.

**Figure 10 sensors-19-00685-f010:**
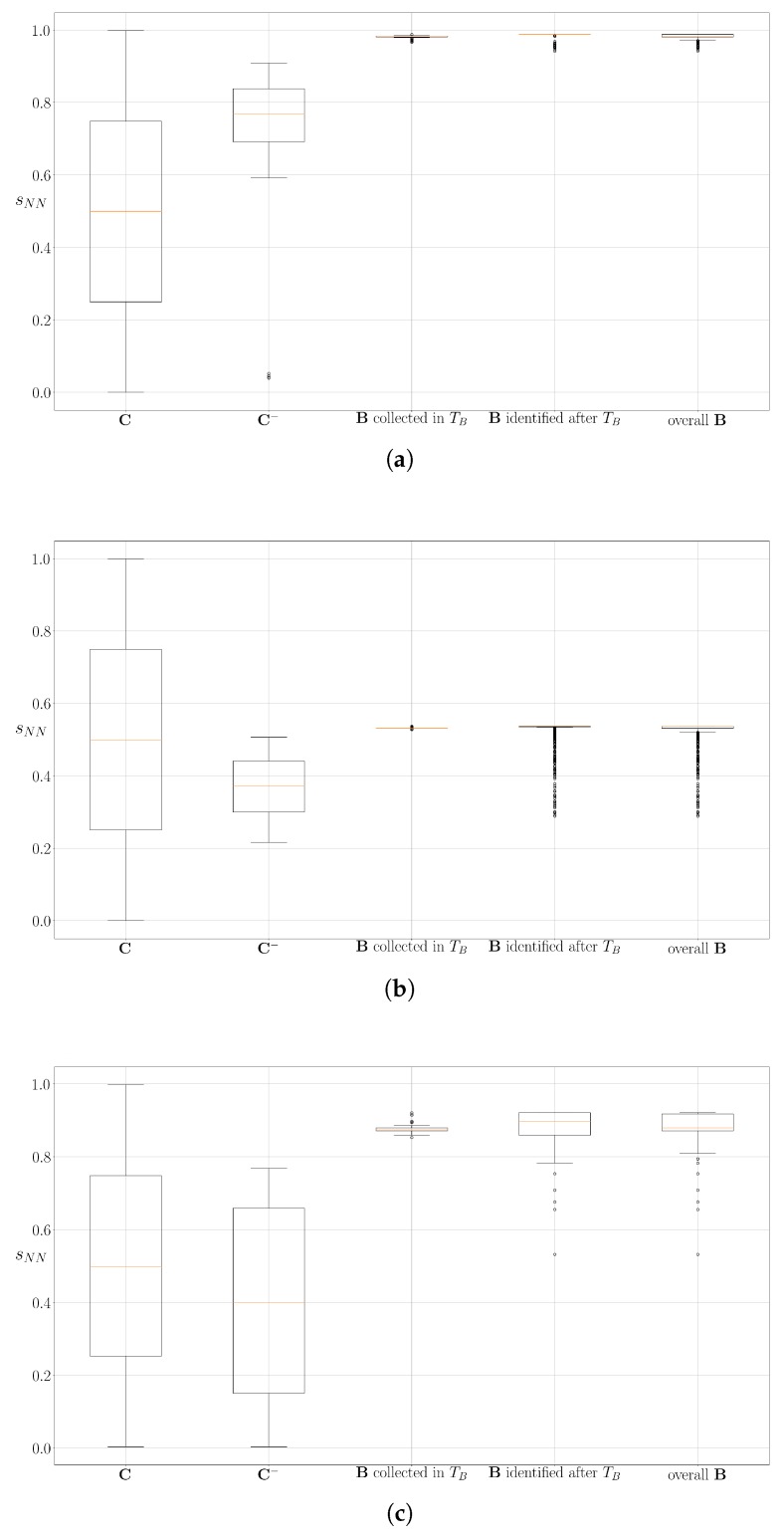
The distributions of OCNN scores ($s_{N}N$) of baseline responses and the measurements in the first exposure state in three experiments, namely (**a**): Experiment 1, (**b**): Experiment 2: and (**c**): Experiment 3. In each subfigure, the first box-plot corresponds to the training set of OCNN (C), which is also the measurements in the first exposure state; The second box-plot corresponds to the data with Top 25%
$s_{GM}$ score in C. This set of measurements is denoted as $C^{-}$; The third box-plot corresponds to the baseline responses collected during $T_{B}$; The fourth box-plot corresponds to the identified baseline responses after $T_{B}$; The fifth box-plot corresponds to all data labeled as baseline responses. In all three experiments shown here, the $s_{NN}$ values of the baseline responses are distributed over a very narrow interval, and this interval does not overlap with $C^{-}$.

**Figure 11 sensors-19-00685-f011:**
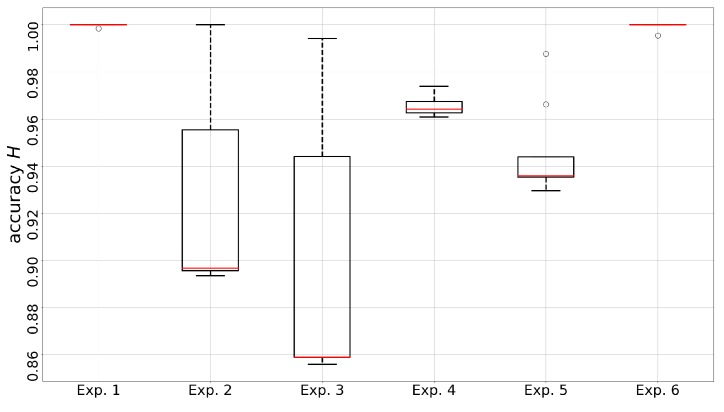
The distributions of accuracy rate *H* of the multi-compound discrimination models in each trial. The discrimination models are periodically updated ($T_{UD}=30$ s) during the exposure state, which may result in different performance.

**Figure 12 sensors-19-00685-f012:**
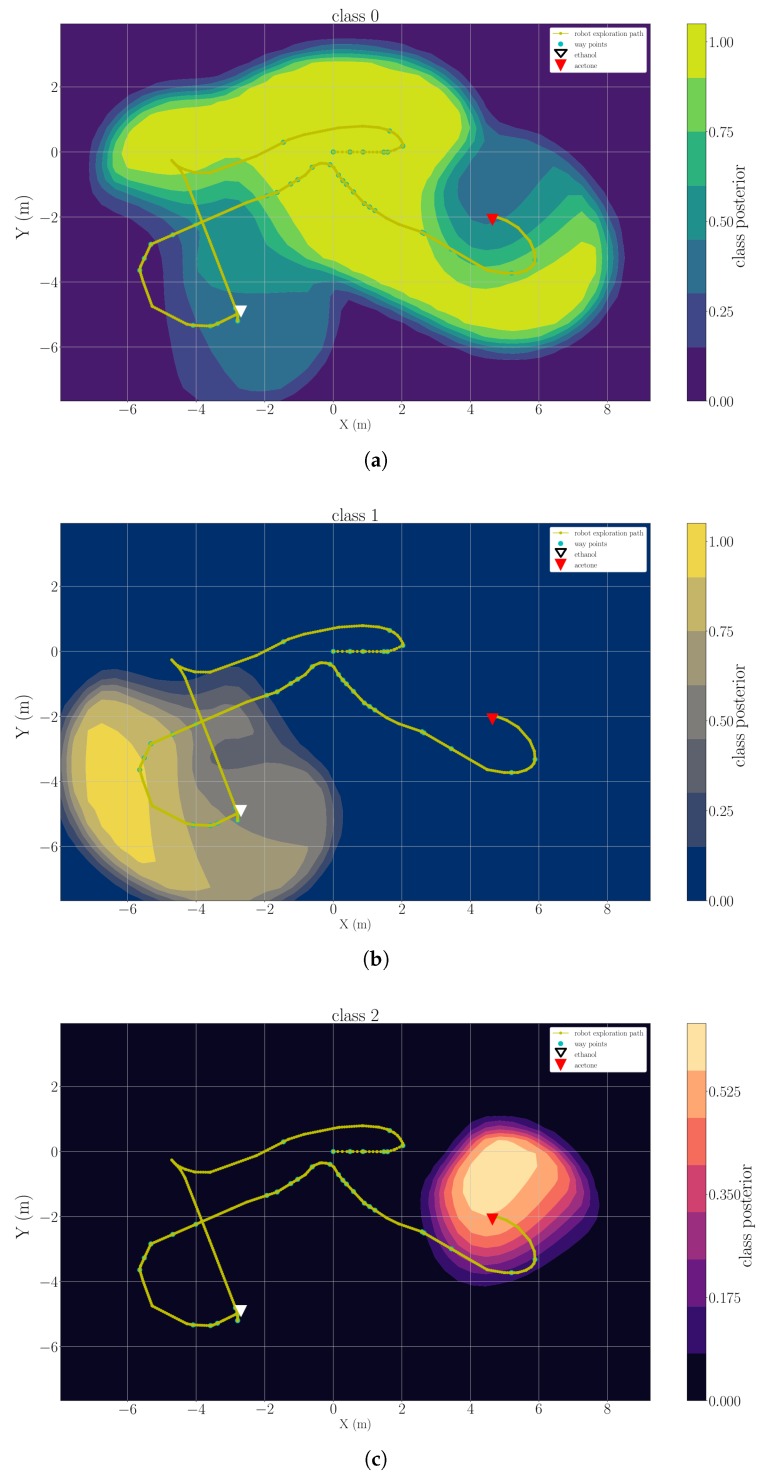
A snapshot of the classification map learned for clean air (**a**), ethanol (**b**) and acetone (**c**) in the trial Experiment 1.

**Figure 13 sensors-19-00685-f013:**
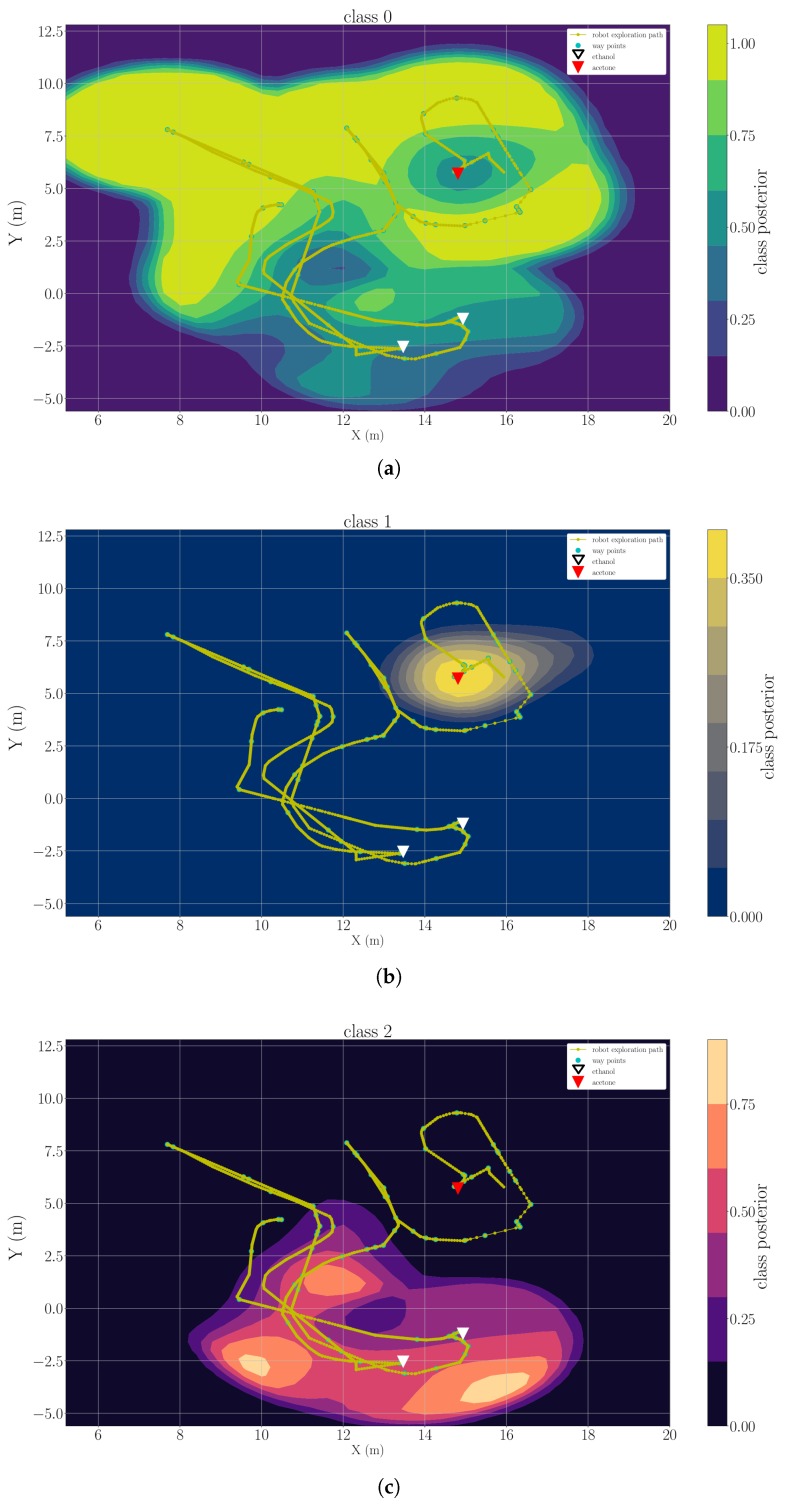
A snapshot of the classification map learned for clean air (**a**), ethanol (**b**) and acetone (**c**) in the trial Experiment 2.

**Figure 14 sensors-19-00685-f014:**
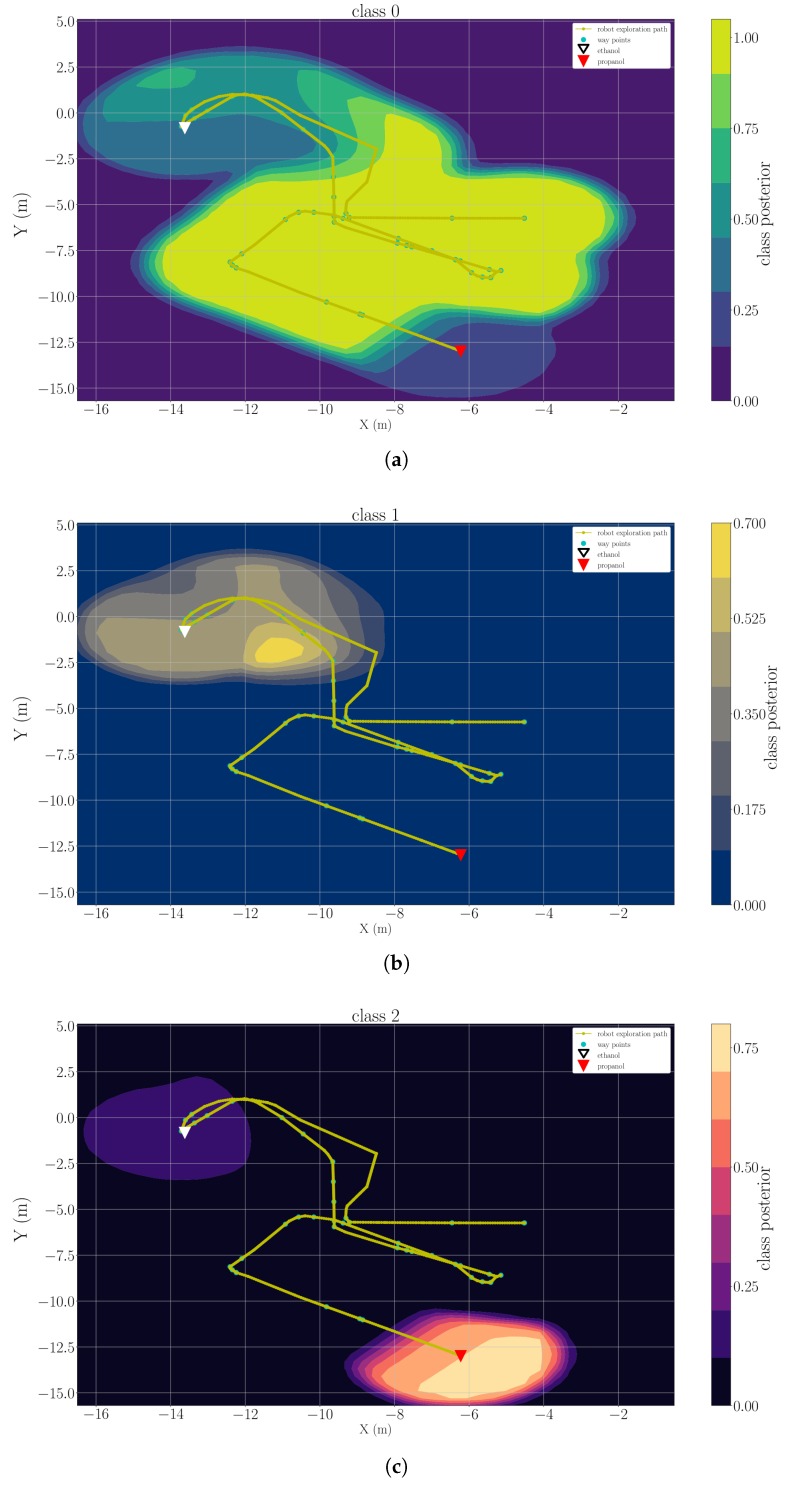
A snapshot of the classification map learned for clean air (**a**), ethanol (**b**) and acetone (**c**) in the trial Experiment 3.

**Figure 15 sensors-19-00685-f015:**
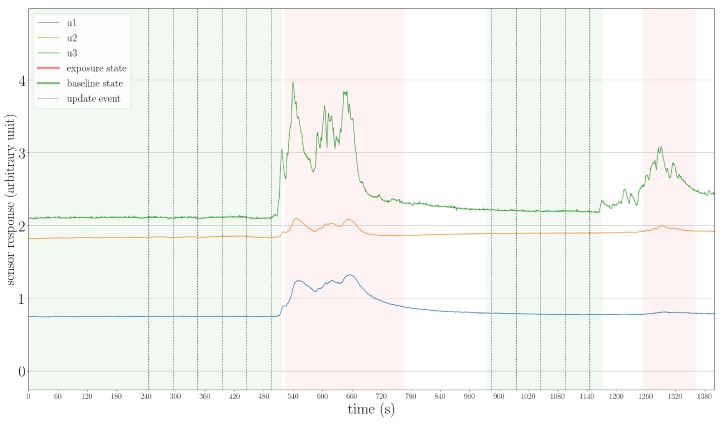
The baseline and exposure states found in Experiment 1. Before the second determined exposure state, some gas detection events are missed by the proposed gas detection module.

**Table 1 sensors-19-00685-t001:** Summary of experiments.

Trial	Duration (min)	Gas Source 1	Gas Source 2	Distance between the Sources (m)
Experiment 1	23	ethanol	acetone	7.8572
Experiment 2	43	(1a) ethanol (1b) ethanol	acetone	1a Vs. 1b: 1.9805 1a Vs. 2: 6.9148 1b Vs. 2: 8.3554
Experiment 3	21	ethanol	propanol	14.2420
Experiment 4	26	ethanol	propanol	12.4036
Experiment 5	18	ethanol	propanol	14.6000
Experiment 6	20	ethanol	propanol	12.7425

**Table 2 sensors-19-00685-t002:** Efficiency comparison between the online and offline KmP gas discrimination approach.

Trial	Data Size Reduction		Final Online $K^{*}$ Learning		Final Offline $K^{*}$ Learning	
	$\mu_{\mathrm{DR}}$ (%)	$\sigma_{\mathrm{DR}}$ (%)	Search Space	$K=K^{*}$	Default Search Space	$K=K^{*}$
Experiment 1	79.19	7.29	4	True	5	True
Experiment 2	72.13	10.06	2	True	5	False
Experiment 3	69.44	11.42	2	True	5	True
Experiment 4	54.04	13.36	3	True	5	True
Experiment 5	66.23	11.43	2	True	5	True
Experiment 6	68.69	10.11	3	True	5	False

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
