# Peer review of "Towards Gas Discrimination and Mapping in Emergency Response Scenarios Using a Mobile Robot with an Electronic Nose"

_sensors, 2019, doi:10.3390/s19030685_

Round 1

Reviewer 1 Report

This manuscript describes a novel e-Nose loaded mobile robot which can be used for gas detection and mapping in emergency response scenarios. The authors focus on gas sensing in harsh environments where the learning and discrimination of the sensor system should be carried out in real time. An ensemble one-class classifier is proposed to solve the problem of computational power in gas discrimination. Also, an unsupervised approach is proved having high accuracy in predicting the number of the chemical analytes. By integrating a discrimination model with a gas distribution mapping algorithm, the system can be used to interpret the environment in the form of classification maps. 

This is well written, high quality, and useful contribution, which may be very significant in the field of mobile robot olfaction. The manuscript can be accepted with its present form. 

Author Response

Thanks for the positive encouragement for the reviewer.

We streamlined the whole document to improve the language quality, and made some changes to response to another reviewer. All the changes are highlighted.

Reviewer 2 Report

This paper proposes a mobile robotic system for gas distribution mapping. An interesting result is that the robot performs online gas discrimination and mapping of multiple chemical sources in an unsupervised way. Nevertheless, in its current version, the paper suffers from drawbacks that may prevent its publication. The system is based on a mix of different techniques (OCGM, OCNN, KmP, Kernel DM+V…) which are described in the paper with great technical details. This level of description makes the paper unclear and difficult to read. Such algorithmic details are not needed as most, if not all, of the techniques have been published elsewhere by the same research group. For the sake of publication, the authors may consider focusing their manuscript on the robotic experiments instead of algorithmic details. They may also consider concluding the manuscript by commenting on the limitations of their approach. For example, in experiment 2, the two ethanol sources are close to each other and are thus considered by the system to be emitted from the same point source (they belong to the same class). Would the result have been similar with two different chemicals at close locations (eg. ethanol and acetone)? Actually, what is the minimum spatial distance for the system to consider separate sources and different classes?

Author Response

The reviewer pointed out two major issues to be addressed. The first is to focus on the robotic experiment and make the description of the methodology easier for readers to follow.  The second one is that the limitation of the proposed gas sensing system should be discussed. Our responses (along with the revised manuscript) are as follows:

 As suggested, we rephrased and removed some content in Section 3, where the detailed introduction of the gas distribution mapping algorithm has been removed.  In Section 4 on page 15 and 16, we added some background information of the robotic experiment. In Section 5, we rephrased some content to better describe the results of the experiments.  These modifications can be found in Section 5.2. and Section 5.3.

The reviewer also pointed out an important limitation of discrimination and mapping multiple chemical compounds using a MOX sensor array.  In Section 6 on page 21 and 25, we added a paragraph of discussion on the causes of such limitation, and under which condition this limitation would affect the performance of the proposed gas sensing system.

We streamlined the whole document to improve the language quality.

Thanks for the helpful and informative review.

Round 2

Reviewer 2 Report

This new version is improved. The authors have taken all my comments into consideration.